# Preparation and Characterization of Salt-Mediated Injectable Thermosensitive Chitosan/Pectin Hydrogels for Cell Embedding and Culturing

**DOI:** 10.3390/polym13162674

**Published:** 2021-08-10

**Authors:** Giulia Morello, Alessandro Polini, Francesca Scalera, Riccardo Rizzo, Giuseppe Gigli, Francesca Gervaso

**Affiliations:** 1Dipartimento di Matematica e Fisica E. De Giorgi, University of Salento, Campus Ecotekne, via Monteroni, 73100 Lecce, Italy; giulia.morello@unisalento.it (G.M.); giuseppe.gigli@unisalento.it (G.G.); 2CNR NANOTEC—Institute of Nanotechnology c/o Campus Ecotekne, via Monteroni, 73100 Lecce, Italy; francesca.scalera@nanotec.cnr.it (F.S.); riccardo.rizzo@nanotec.cnr.it (R.R.)

**Keywords:** natural polymers, thermoresponsive hydrogels, semi-IPN system, 3D in vitro models, cell encapsulation

## Abstract

In recent years, growing attention has been directed to the development of 3D in vitro tissue models for the study of the physiopathological mechanisms behind organ functioning and diseases. Hydrogels, acting as 3D supporting architectures, allow cells to organize spatially more closely to what they physiologically experience in vivo. In this scenario, natural polymer hybrid hydrogels display marked biocompatibility and versatility, representing valid biomaterials for 3D in vitro studies. Here, thermosensitive injectable hydrogels constituted by chitosan and pectin were designed. We exploited the feature of chitosan to thermally undergo sol–gel transition upon the addition of salts, forming a compound that incorporates pectin into a semi-interpenetrating polymer network (semi-IPN). Three salt solutions were tested, namely, beta-glycerophosphate (βGP), phosphate buffer (PB) and sodium hydrogen carbonate (SHC). The hydrogel formulations *(i)* were injectable at room temperature, *(ii)* gelled at 37 °C and *(iii)* presented a physiological pH, suitable for cell encapsulation. Hydrogels were stable in culture conditions, were able to retain a high water amount and displayed an open and highly interconnected porosity and suitable mechanical properties, with Young’s modulus values in the range of soft biological tissues. The developed chitosan/pectin system can be successfully used as a 3D in vitro platform for studying tissue physiopathology.

## 1. Introduction

In recent years, scientific research has focused its attention on the development of three-dimensional (3D) in vitro tissue models, used in various research areas such as drug development and precision medicine [1,2]. Three-dimensional culture models, mimicking biological tissues’ architecture and microenvironment, can be very useful for studying several disease-related mechanisms such as tumor onset, progression and metastasis [3]. Ideally, a 3D tissue model should faithfully reproduce the typical cellular architecture of the native tissue and provide a network of molecules that play a key role in disease progression [4,5]. The importance of suitable 3D in vitro models is strictly linked to the need to bridge the gap between traditional 2D cell cultures and animal models used in scientific research [6]. Although 2D cultures have been and are still widely employed to study various diseases, they have a poor ability in reproducing the physiological tissue architecture [7]. On the other hand, although animal models have undoubted advantages over in vitro cultures, the experimental costs, ethical concerns and species variability often do not allow for the extension of therapies to humans [8]. In order to obtain physiologically relevant 3D systems to be successfully used to study pathological mechanisms, hydrogels are promising candidates thanks to their versatility and tailoring possibilities. Polymeric hydrogels are cross-linked macromolecular networks widely used for several biomedical applications, ranging from tissue engineering to drug delivery [9]. In 3D in vitro cultures, hydrogels facilitate the interaction and exchange of metabolites between cells and the matrix [10] and, acting as a “scaffold,” allow cells to self-organize in vitro similarly to what occurs in vivo. Hydrogel systems can be classified into natural, synthetic or hybrid hydrogels depending on their composition. Natural hydrogels are biocompatible, biodegradable and, being very similar to the native tissue extracellular matrix (ECM) from a physiochemical point of view, able to interact with living cells, while synthetic systems are generally inert, are not biodegradable and, consequently, do not allow a proper matrix remodeling [11]. Natural hydrogels include polysaccharides, proteins and animal derivatives, all being extensively used for cell culture studies. However, their low mechanical properties and poor stability, generally related to the physical cross-linking mechanism, limit their use for long-term culture [12]. In order to improve the performance of single-polymer hydrogels, hybrid hydrogels, composed of two different polymers, can be developed that present more controlled physicochemical properties [13]. Among hybrid hydrogels made of natural polymers, chitosan/pectin systems have recently been proposed for tissue engineering applications and drug delivery systems [14,15]. Chitosan, a deacetylated derivative of chitin, is a cationic polysaccharide, localized in the exoskeleton of crustaceans [16]. It is highly biodegradable and biocompatible and also shows interesting wound-healing, anti-microbial and anti-tumor properties [17,18,19]. Pectin, a linear, non-toxic, anionic plant polysaccharide, is characterized by the presence of galacturonic acid molecules and is widely used for biomedical applications [20,21]. Though Pec/Ch systems have already been proposed for cell culture, their combined use to prepare an injectable thermosensitive hydrogel suitable for cell embedding and culturing has not been reported yet.

As reported in the literature, chitosan interacts with salt solutions, such as beta-glycerophosphate (βGP), for obtaining thermosensitive hydrogels at physiological conditions [22,23,24]. In this study, we exploited such Ch feature to form a hybrid compound by incorporating a second polymer, pectin, allowing greater flexibility without altering the system stability. Three weak bases were tested, namely, βGP, phosphate buffer (PB) and sodium hydrogen carbonate (SHC), already used in the literature for their ability to raise the pH of acidic Ch solutions up to physiological values and induce a sol–gel transition at the physiological temperature [22,25]. All the obtained hydrogels were characterized in terms of pH measurement, injectability, thermosensitive sol–gel transition, swelling properties, in vitro stability, morphology and cell viability. Rheometer analysis was performed to evaluate the rheological properties of the system, and FT-IR analysis was carried out to outline the nature of the interactions between the polymer chains and the gelling agents. Finally, the potential of the hydrogels as 3D in vitro systems for cell embedding and culturing was also investigated by preliminary biological tests, performed to evaluate the viability of a colon rectal cancer cell line (HCT 116) embedded into the systems.

## 2. Materials and Methods

### 2.1. Preparation of Chitosan/Pectin Hybrid Hydrogels

Low-molecular weight chitosan (Ch) (#448869, Sigma Aldrich, Milan, Italy) and pectin (Pec) from citrus peel (galacturonic acid ≥74.0% dried basis) (#P9135, Sigma Aldrich, Milan, Italy) were used for hydrogel preparation. Hydrogels were prepared by solubilizing Ch powder 3.33% (*w*/*v*) in 0.1 M hydrochloric acid (HCl) solution and Pec powder 3.33% (*w*/*v*) in ultrapure Milli-Q water under stirring at room temperature (r.t.) overnight. The concentration of the starting polymer solution was selected after preliminary tests, aimed at obtaining the most concentrated polymer (Ch) solution by stirring at room temperature (i.e., avoiding high temperatures) and using a low HCl molarity. The gelling agent (GA) solutions were prepared by dissolving βGP, PB and SHC powders in Milli-Q water to a final concentration of 0.04 M (see Table 1). The two polymer solutions, previously centrifuged at 1500 rpm for 5 min at 4 °C, were mixed in an optimized ratio (50:50), and the resulting mix Ch-Pec with pH 6.00 was stored at 4 °C until use. For hydrogel preparation, the three GA solutions, βGP, PB and SHC, were alternately added to the Ch-Pec mix in the optimized volume ratio of 5:1 (Ch-Pec mix/GA solution) using two syringes joined by a Luer-lock connector and then centrifuged at 1500 rpm for 2 min at r.t. The pre-hydrogel solutions were then incubated at 37 °C for 2 h, in order to thermally induce the sol–gel transition, with or without the drop-by-drop addition of 500 µL of Dulbecco’s Modified Eagle Medium (DMEM) to simulate cell encapsulation.

### 2.2. pH Measurement, Injectability and Inversion Tube Test

The pH of Ch, Pec and GA solutions, the Ch-Pec mix and the final hydrogels was measured by test strips and monitored during the gelation process. Injection tests were performed by injecting all the hydrogel samples through a syringe equipped with a 23G needle. The behavior of the injected solutions was monitored by visual inspection. In order to evaluate the thermosensitive sol–gel transition during the inversion tube test, the solutions were injected in vials, and the fluidity/viscosity of the hydrogels was visually assessed through the inversion of the vial, at r.t. and at different time points at 37 °C.

### 2.3. Rheological Analysis

Rheological tests on Ch-Pec-βGP, Ch-Pec-PB and Ch-Pec-SHC formulations were performed using an Anton Paar instrument (Physica MCR 301, Ostfildern, Germany) equipped with a two-plate geometry (plate diameter 25 mm, gap distance 0.5 mm) and connected to a circulating water bath. Immediately following the preparation of hydrogel samples, the variation in the storage modulus (G′) and loss modulus (G”) with temperature was measured, at a constant shear strain (5%) and frequency (1 Hz). The temperature was increased from 5 to 50 °C at a rate of 1 °C/min, and the solutions were kept at 4 °C before mixing. Each test was performed in duplicate.

### 2.4. Fourier Transform Infrared (FTIR) Spectra

FTIR spectra were obtained to understand the molecular interactions and functional group characterization. The FTIR spectra were obtained using an FT/IR-6300 type A spectrophotometer (JASCO, Easton, MD, USA) in ATR-FTIR mode. All spectra were recorded with the resolution of 4 cm^−1^ in the range of 400–4000 cm^−1^ with 100 scans. The measurements were performed on Ch, Pec, Ch-Pec, Ch-Pec-βGP, Ch-Pec-PB and Ch-Pec-SHC freeze-dried hydrogels in order to evaluate the interactions of the Ch amino groups with Pec, βGP, PB and SHC.

### 2.5. Swelling Test and In Vitro Stability

The swelling ability of the hydrogels was assessed through gravimetric measurements. Briefly, after 2 h of incubation at 37 °C, samples were frozen at −20 °C and then lyophilized overnight (LIO 5P, Cinquepascal, Milan, Italy). The weight of samples was recorded using an analytical balance at the dry state, immediately after hydration in phosphate-buffered saline (PBS) and at different time points at 37 °C. The swelling ratio percentage (SR) was calculated according to the following Formula (1), where W_dry_ is the initial dry weight of the hydrogel, and W_wet_ is the weight of the hydrogel after hydration in PBS and incubation at 37 °C:SR (%) = [(W_wet_ − W_dry_)/W_dry_] × 100(1)

The non-enzymatic degradation of the hydrogel over time was evaluated through the stability test. After 2 h of incubation at 37 °C, samples were weighed (t = 0), and the weight was then monitored at different incubation times at 37 °C in PBS. The percentage of weight loss (WL) was calculated according to the following Formula (2), where W_0_ is the initial weight of the hydrogel at t = 0 after thermal gelation at 37 °C, and W_i_ is the weight of the hydrogel after its hydration in PBS at 37 °C at the different selected time points:WL (%) = [(W_0_ − W_i_)/W_0_] × 100(2)

### 2.6. Morphological Analysis

The porous structure of the hydrogels was observed by scanning electron microscopy (SEM) (Zeiss Sigma 300 VP FE-SEM, Carl Zeiss AG, Oberkochen, Germany). After 2 h of incubation at 37 °C, samples were frozen for at least 2 h at −20 °C and then freeze dried overnight. Samples were sectioned, gold sputtered and then observed under an SEM microscope at different magnifications. Finally, the measurement of the hydrogel pore diameter was statistically evaluated using ImageJ software (ImageJ bundled with 64-bit Java 1.8.0_172, NIH). Five SEM images (magnification 5×) were analyzed for Ch-Pec-βGP, Ch-Pec-PB and Ch-Pec-SHC samples, measuring the diameter as an average of two measurements for each pore, with a total of approximately 250 pores per sample.

### 2.7. Compression Test

To evaluate the hydrogel stiffness, hydrogel samples underwent an unconfined compression test. Briefly, samples were tested after 2 h of incubation at 37 °C; initial diameter and thickness were recorded, and then the sample was loaded between two impermeable and non-lubricated compression plates and tested in compression in “wet” conditions at r.t. using a universal uniaxial machine (ZwickiLine 1kN, Zwick Roell, Kennesaw, GA, USA), equipped with a 10 N load cell, up to 40% deformation and with a displacement velocity of 2 mm/min. The average Young modulus (E) was calculated as the slope of the linear part of the stress–strain curves at low strain values (0–5%) for each hydrogel formulation with and without DMEM. The respective mean values were compared with each other (n = 4).

### 2.8. Cell Culture

Colorectal carcinoma cells (HCT 116, ATCC CCL-247, LGC Standards, Milan, Italy) were cultured in DMEM with 4.5 gL^−1^ glucose and sodium pyruvate without L-glutamine supplemented with 2 mM L-glutamine, 10% Fetal Bovine Serum (FBS), 100 U mL^−1^ penicillin and 100 µg mL^−1^ streptomycin. Cells were incubated at 37 °C with 95% of humidity and 5% of carbon dioxide (CO_2_), and 0.05% Trypsin-Ethylenediaminetetraacetic acid (Trypsin-EDTA 1X) was used regularly to pass cells every 2–3 days until 90% confluence was reached.

### 2.9. Analysis of Cell Encapsulation in Hydrogels by Nucleus Staining

HCT 116 cell encapsulation in the hydrogel formulations (βGP, PB, SHC) was evaluated using a Hoechst 33342 fluorescent stain for nuclei (NucBlue Live ReadyProbes Reagent, Thermo Fischer Scientific, Monza, Italy). Briefly, 3 mL of complete DMEM with 6 drops of reagent solution was prepared. After preliminary tests, a density of 2 million cells per mL of hydrogel was chosen for encapsulation, by gently mixing the cell suspension into the hydrogel solution. Two million cells, resuspended in 166 μL of complete DMEM, were encapsulated in 1 mL of hydrogel. Then, 100 µL spots of HCT 116 cell-embedded hydrogel were incubated at 37 °C 5% CO_2_. Cell encapsulation within the hydrogels and early biocompatibility assessment were evaluated after 24 h according to the following protocol. Briefly, once the encapsulation was carried out, 1 mL of DMEM distributed in the wells of a multiwell was removed and washed with PBS, and 300 µL of DMEM reagent was introduced. The samples were incubated for 2 h at 37 °C and observed under a fluorescence microscope (EVOS M7000, Thermo Fisher Scientific, Monza, Italy). Similarly, we assessed the late biocompatibility of the hydrogel systems after 21 days of culture by optical microscopy. Z-stack analysis was performed on Ch-Pec-βGP samples at different magnifications (196 slices at 10×, 92 slices at 40×), with a step size of 2 µm.

### 2.10. Statistical Analysis

All experiments, unless differently specified, were performed in triplicate, and the results are reported as the mean ± standard deviation. Data analysis and graphing were performed with Microsoft Excel 2019. Regarding the compression tests, GraphPad Prism software (v. 8.4.2) was employed to perform statistical analysis, using one-way ANOVA analysis.

## 3. Results

In the present study, a novel injectable thermosensitive hybrid hydrogel with Ch and Pec was developed (Figure 1). Three weak bases, namely, βGP, PB and SHC, were tested as GA. All of them, once added to the Ch-Pec mix, allowed forming a stable chitosan network incorporating Pec inside at 37 °C, thus originating a semi-interpenetrating polymer network (semi-IPN). Preliminary tests (data not shown) allowed selecting the best GA concentrations able to increase the acidic pH of the Ch-Pec mix to the physiological value of 7.4 and to induce the sol–gel transition at 37 °C. The different hydrogel formulations were physicochemically characterized, and a preliminary biological characterization was performed to assess the capability of the systems to embed cells for 3D in vitro culture.

### 3.1. pH Measurement, Injectability and Inversion Tube Test

The three GA solutions, βGP, PB and SHC, were alternately added in appropriate and optimized concentrations to the Ch-Pec mix, and the different formulations tested, with or without the addition of cell culture medium (DMEM), are shown in Appendix A. In Table 1, the final polymer and GA solution concentration and the pH values of gelling agents, initial polymer solutions and final hydrogels after 2 h at 37 °C are reported. All hydrogel formulations, despite the initial acidic pH values of the polymer solutions, reached a pH value of 7.4 immediately after the GA and DMEM addition, which was used to simulate cell encapsulation. A summary of three hydrogel formulations’ behavior in terms of injectability and sol–gel transition is shown in Appendix A. All prepared hydrogel formulations were injectable at r.t., through a G23 needle (Appendix A). To evaluate the thermally induced sol–gel transition, inversion tube tests were performed (Appendix A). Although the three formulations could not easily flow at r.t. because of their high viscosity, the increase in the temperature to a physiological value was a sine qua non to induce the sol–gel transition. Indeed, without incubation for 2 h at 37 °C, a stable gel state could not be achieved, confirming that gelation was temperature-mediated.

### 3.2. Rheological Analysis

The rheological properties of the Ch-Pec-βGP, Ch-Pec-PB and Ch-Pec-SHC hydrogel formulations were studied by heating samples from 5 to 50 °C at a rate of 1 °C/min, at a constant shear strain and frequency. The temperature dependence of the hydrogel storage modulus (G′) and loss modulus (G″) is reported in Figure 2. Upon heating from 5 to 50 °C, the temperature at which G′ and G″ rapidly increase and the slope of this increase provide an indication of the temperature of the incipient gelation [23]. As it can be observed in the diagrams in Figure 2, G′ and G″ increased rapidly in all three hydrogel formulations at a temperature of about 37 °C. In Ch-Pec-βGP hydrogels, the two moduli present an abrupt increase at the physiological temperature that is, conversely, less sharp and sudden in the Ch-Pec-PB and Ch-Pec-SHC systems, which present a broader temperature gelation range.

### 3.3. FT-IR Analysis

Figure 3A shows IR spectra of Ch, Pec and Ch–Pec mixing. All spectra exhibit a strong and broad nonsymmetric band at about 3430 cm^−1^ that results from the overlapping of the O-H and N-H stretching vibrations of the functional groups engaged in the hydrogen bonds. Two bands at 1740 and 1610 cm^−1^ in the pectin spectrum are attributed to esterified and nonesterified carboxyl groups, respectively [15]. Characteristic peaks of chitosan are observed at the 1633 cm^−1^ peak of amide I (C=O band), and at the 1535 cm^−1^ amide II band [26]. When chitosan and pectin were mixed, shifting to lower wavenumber values for amide I (1624 cm^−1^) and amino groups (1526 cm^−1^) was detected in the spectrum of Ch-Pec. Further slight variation in the stretching frequency was noticed upon the addition of salts into the system (Figure 3B).

### 3.4. Swelling Test and In Vitro Stability

Swelling and stability tests were performed on the hydrogel prepared using the three different GAs, with or without the addition of DMEM. The swelling test up to 21 days of incubation at 37 °C (Figure 4A–D) shows that hydrogels present a high swelling capacity already in the first 10 min of incubation in PBS at 37 °C. The βGP formulations, with and without DMEM, present a comparable trend and a swelling ratio of around 2000%. The sample with PB shows a lower swelling capacity, which increases with the addition of DMEM. On the contrary, in the SHC sample, the addition of DMEM reduces the swelling ability. Generally, all samples were able to retain a high amount of water, reaching the equilibrium very fast and remaining stable for up to three weeks, except for the formulation with PB+DMEM (Figure 4D), where the degradation process began at day 7, in accordance with the stability test results.

The stability test was performed on different hydrogel formulations with or without DMEM up to 25 days of incubation at 37 °C, as shown in (Figure 5A,B). All systems were stable in weight up to 7 days, and the addition of DMEM induced a decrease in the sample stability, promoting a rapid sample degradation in some cases (such as PB+DMEM and SHC+DMEM). Among all the formulations, the βGP sample was the most stable over time, showing the best swelling ability.

### 3.5. Morphological Analysis

The structure of the different hydrogel formulations, with and without DMEM addition, was studied by SEM and optical microscopy. As shown in Figure 6 and Appendix A, the morphological analysis at different magnifications showed that all samples presented an open and highly interconnected pore structure. The analysis of the acquired images (Figure 7) allowed estimating the pore diameter, whose average value was about 220 µm, with comparable values among the three formulations. The βGP and SHC formulations showed similar results between the +DMEM and −DMEM formulations, while PB samples showed a slight decrease in pore diameter, though not significant, in the formula with DMEM.

### 3.6. Compression Test

The mechanical properties of the hydrogels were evaluated through compression tests, which allowed observing that there were no significant differences between the formulations of hydrogels with and without the addition of DMEM, indicating that the amount of DMEM used was actually small enough to not alter the mechanical properties of the system. The compression test showed a very low Young modulus (between 1 and 2 kPa) for all the different samples without any significant difference among the hydrogel types (Figure 8).

### 3.7. Cell Encapsulation in Hydrogels by Nucleus Staining

To evaluate the morphology and encapsulation of HCT 116 cells throughout the samples, the cell-laden hydrogels were stained for nuclei after 24 h of culture and observed by fluorescence microscopy (Figure 9A). HCT 116 cells spread over the entire volume of all the hydrogels, indicating that this method allowed the homogeneous encapsulation of the cells through the hydrogel samples. However, the cells appeared significantly more numerous in Ch-Pec-βGP than the other two formulations. Cell growth was analyzed after 21 days of culture to assess the late biocompatibility of the hydrogel systems. We noticed the formation of cell aggregates (spheroids) in all the formulations (Figure 9B). Overall, Ch-Pec-βGP displayed a higher number of spheroids throughout the sample (Figure 9C, Appendix A).

## 4. Discussion

In this study, we aimed at developing a chitosan/pectin hydrogel system suitable for cell embedding and culturing, meeting, therefore, some fundamental chemico-physical requirements, such as (i) a physiological pH, (ii) injectability at r.t. and (iii) ability to gel at the physiological temperature, i.e., 37 °C. Several hydrogels with chitosan and pectin have been proposed in the literature; however, they are usually in the gel state at r.t. and in the sol state at high temperatures used to solubilize the two polymers (60 to 97 °C), conditions not suitable for cell viability [14,15,16,27,28,29,30,31,32,33,34]. To overcome the limits of the chitosan/pectin systems reported in the literature, not compatible with cell embedding applications, in the present study, we decided to exploit the well-known ability of chitosan solution to gel at 37 °C thanks to the addition of weak bases such as βGP, in order to induce the formation of a chitosan hydrogel network that incorporates pectin inside, giving rise to a semi-IPN. Here, besides βGP, PB and SHC were also tested as potential gelling agents able to induce the thermal sol–gel transition of the Ch-Pec mix. All the systems (βGP, PB and SHC) presented a pH suitable for cell viability, reaching a pH value of 7.4 immediately after mixing with GAs, resulting in being injectable at r.t., with a sol–gel transition at 37 °C. Although they were injectable, in the tube inversion test, all the formulations could not easily flow already at r.t. due to their high viscosity. However, a stable gel state could be achieved only by increasing the temperature to 37 °C, demonstrating that without heating, gelling did not occur. Moreover, the thermosensitive behavior of the hydrogel was confirmed by rheological analysis performed on the Ch-Pec mix immediately after the salt solution addition. The rheological results highlight the salt-mediated thermosensitive gelation of the systems, showing a sudden and abrupt increase in the storage and loss moduli of all the hydrogel formulations. The Ch-Pec-βGP formulation took place in the narrowest range of temperature, with respect to the PB and SHC system.

From the FTIR analysis, we witnessed a shifting to lower wavenumber values for amide I (1624 cm^−1^) and amino groups (1526 cm^−1^) in Ch-Pec systems, indicating a change in the surroundings of these groups due to an ionic interaction of protonated amino groups of chitosan and the carboxyl groups of pectin [35,36,37]. With the addition of βGP, PB and SHC salts, a further slight variation in the stretching frequency was observed, likely due to amine group deprotonation [38]. According to Assad et al. [22], in fact, using SHC leads to a neutralization of the Ch chains [39], while using PB or βGP may hinder some NH^2+^ groups within the chain network after the interaction between protonated Ch and negatively charged PB or βGP [25].

The three hydrogels reached very fast (less than 30 min) and very high swelling values (between 1500 and 2000%) in both formulations with and without DMEM, faster and higher values than similar Ch-Pe systems proposed by other authors that reported lower swelling ratio values (about 370% [27], 150–200% [40] or slower [15]) [14,29,31,41]. Regarding the in vitro stability, tested up to 25 days, hydrogels without DMEM were stable up to 25 days, except for the Ch-Pec-PB samples that completely degraded after 7 days. The addition of DMEM, used to simulate the cell delivery within the system, induced a decrease in the sample stability and, in the Ch-Pec-PB-DMEM and Ch-Pec-SHC-DMEM hydrogels, favored a faster degradation. Among all formulations, samples with βGP resulted in being the most stable, and samples with PB resulted in being the weakest. Morphological analysis performed by SEM allowed analyzing the structure of the hydrogels, which presented an open and highly interconnected pore structure and a pore diameter value in the range of 180–250 µm. The Ch-Pec-PB-DMEM formulation showed the smallest pores probably because of the weak structure that partially collapsed.

The performed mechanical characterization by means of compression tests highlighted a very low Young modulus (between 1 and 2 kPa) for all hydrogel formulations tested, a value that falls within the stiffness range of the ECM of soft biological tissues, such as the nervous tissue, whose matrix in healthy conditions presents a stiffness between 0.1 and 1 kPa [42], or the healthy colon tissue, between 2 and 5 kPa [43]. Furthermore, no significant differences were evidenced between the hydrogel formulations with and without the addition of DMEM, indicating that the amount of DMEM introduced into the system to load the cells inside did not induce any change in the mechanical properties of the hydrogel. Our findings are in agreement with those reported by Bombaldi de Souza and colleagues in 2020 [44], where the elastic modulus of chitosan and pectin tubular scaffolds was lower than 2 kPa in the strain range between 5 and 20%, although this value was measured performing a tensile testing test. Unfortunately, to the best of our knowledge, there are no Ch-Pec systems in the literature characterized by compressive tests. However, our values are close to those reported for other hydrogel systems [24,45] and are especially in the range of several human soft tissues [46].

Finally, HCT 116 cells were encapsulated in the different formulations to assess their potential application as cell-embedding hydrogels in 3D in vitro models. Cells were successfully inserted into all the formulations, and the following nuclei staining analysis at 24 h of culture demonstrated the superiority of Ch-Pec-βGP as a cell-embedding system. In fact, in the Ch-Pec-βGP hydrogel, cells appeared numerous and homogenously embedded within the matrix in comparison to the other formulations. This is likely due to the presence of a more stable polymer network facilitated by βGP, as also confirmed by the physicochemical results. However, all the formulations successfully led to the formation of cell aggregates (spheroids) after 21 days of culture. This phenomenon was also reported in other hydrogel systems embedding HCT 116 cells [47,48,49]. These findings highlight the suitability of our systems as an artificial 3D matrix for in vitro models, with the βGP-based formulation confirming its superiority over the others for long-term cell culture.

Overall, the stability and mechanical results highlight that the introduction of Pec within the Ch/βGP system was able to originate a very soft hydrogel compared to the single-polymer network [24] without altering its stability in vitro, two difficultly co-existing characteristics, indicating that such a hybrid polymer system could be an excellent candidate as a 3D ECM analogue of very soft tissues in long-term 3D in vitro culture.

## 5. Conclusions

In the hydrogel systems here proposed, we demonstrated that it is possible to exploit the well-known ability of Ch to form a stable hydrogel network with different gelling agents, namely, βGP, PB and SHC, to incorporate a second polymer within the system, Pec, and originate a salt-mediated thermosensitive Ch-Pec hydrogel. The developed systems reached a sol–gel transition at 37 °C, had a physiological pH compatible with cell embedding, were stable over a long time (25 days in culture conditions), were able to retain a high amount of water and presented mechanical properties in the range of soft biological tissues. Finally, preliminary cell encapsulation tests evidenced the ability of the Ch-Pec-βGP system to host cells that resulted in being homogenously dispersed within the matrix. All the here reported Ch-Pec-βGP hydrogel features make it an ideal candidate as an ECM analogue for long 3D in vitro cell culture.

## Figures and Tables

**Figure 1 polymers-13-02674-f001:**
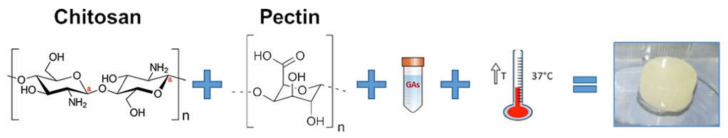
Schematic of the temperature-induced gelation system of chitosan and pectin-based hydrogels.

**Figure 2 polymers-13-02674-f002:**
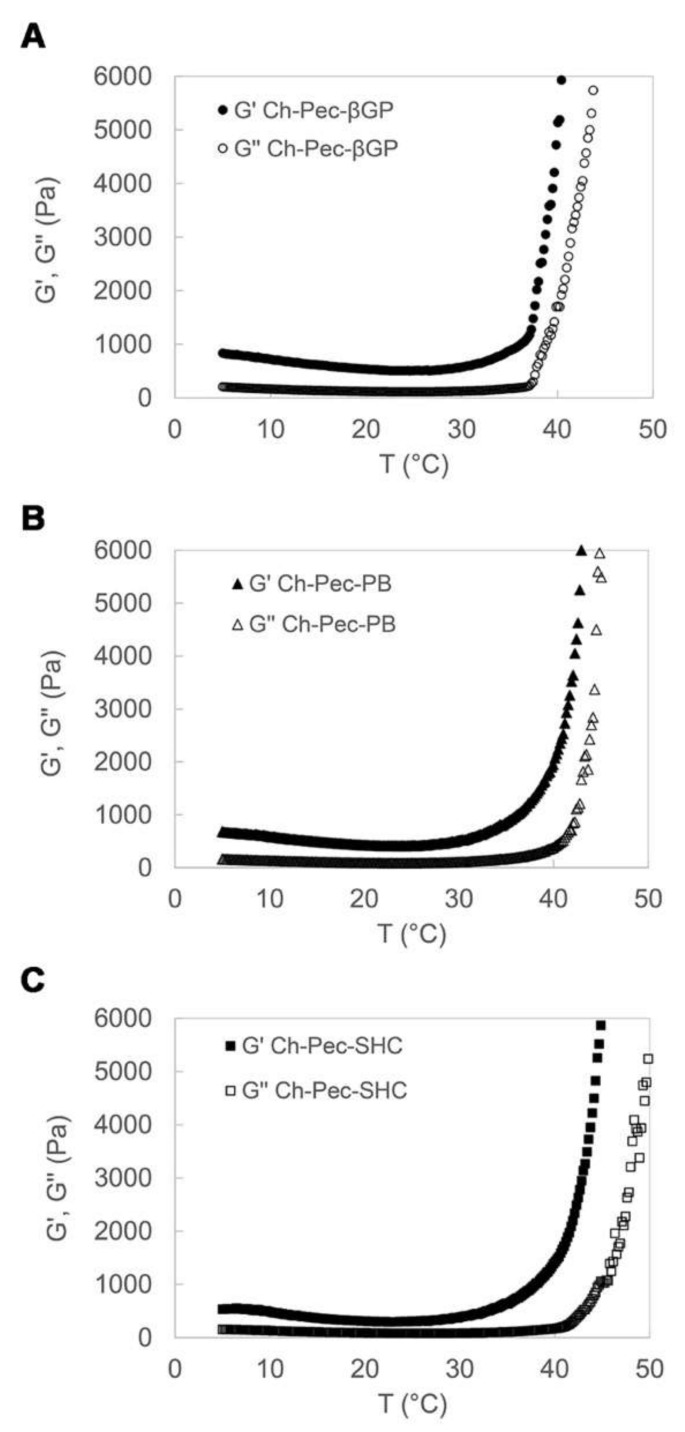
Temperature dependence of Ch-Pec-βGP (**A**), Ch-Pec-PB (**B**) and Ch-Pec-SHC (**C**) hydrogel storage modulus (G′) and loss modulus (G″), upon heating from 5 to 50 °C at a rate of 1 °C/min.

**Figure 3 polymers-13-02674-f003:**
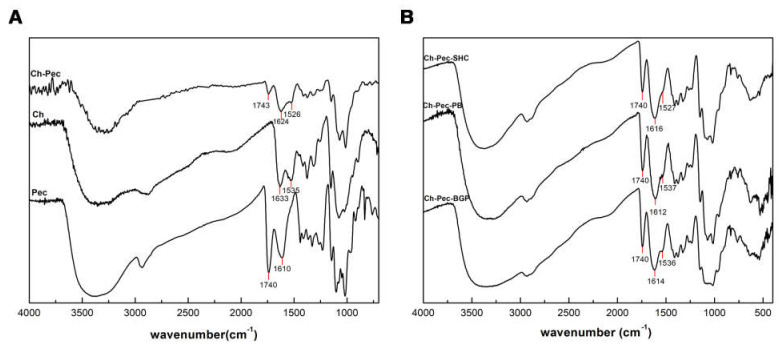
IR spectra of Ch, Pec and Ch–Pec mixing (**A**). IR spectra of Ch–Pec mixing with salts (βGP, PB and SHC) (**B**).

**Figure 4 polymers-13-02674-f004:**
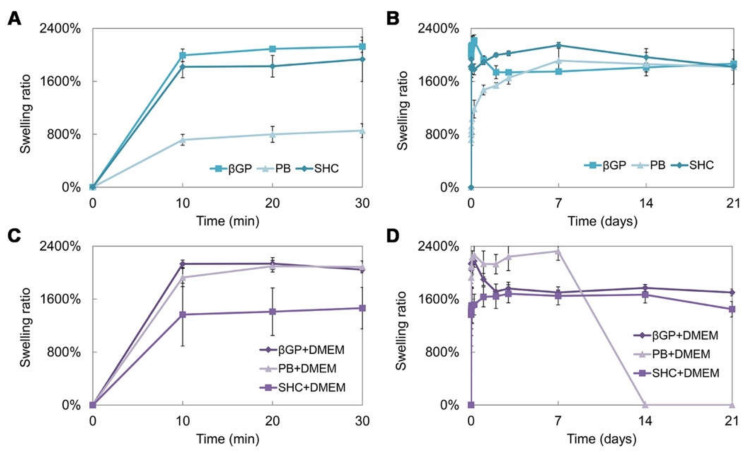
Swelling ratio of the three hydrogel formulations (βGP, PB, SHC) without DMEM in the first 30 min (**A**) and after 21 days of incubation in PBS at 37 °C (**B**), and with DMEM in the first 30 min (**C**) and after 21 days of incubation in PBS at 37 °C (**D**).

**Figure 5 polymers-13-02674-f005:**
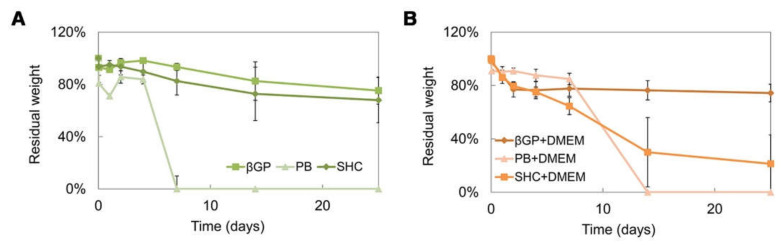
Residual weight percentage after 25 days of incubation in PBS at 37 °C of the three hydrogel formulations (βGP-PB-SHC) without DMEM (**A**), and with DMEM (**B**).

**Figure 6 polymers-13-02674-f006:**
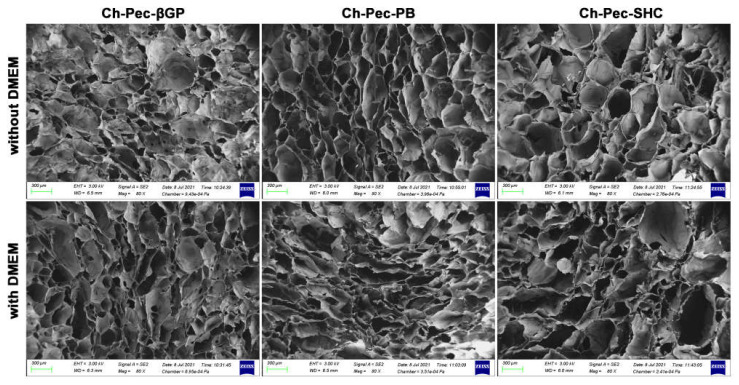
SEM investigation: morphological analysis of hydrogel formulations (βGP-PB-SHC), with and without the addition of DMEM (magnification 80×, scale bar 300 µm).

**Figure 7 polymers-13-02674-f007:**
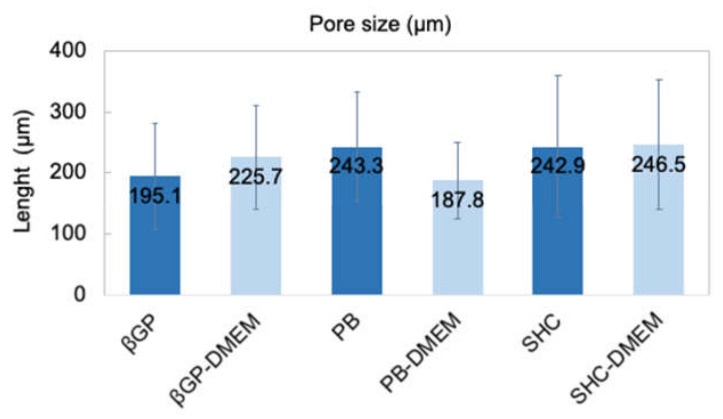
Average pore size of three hydrogel formulations (βGP-PB-SHC) with and without the addition of DMEM.

**Figure 8 polymers-13-02674-f008:**
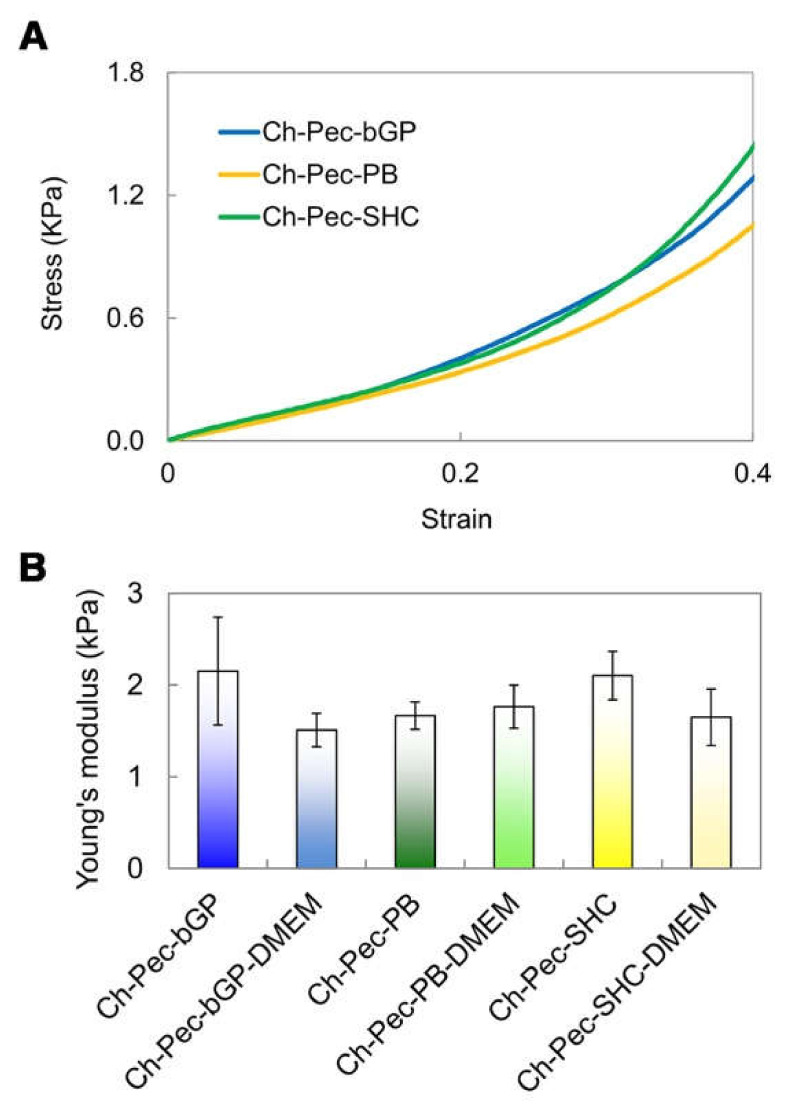
Compression test: (**A**) Stress–strain curves of the hydrogel formulations (βGP-PB-SHC). (**B**) Average values of Young’s modulus for all three hydrogels with and without DMEM (n = 3).

**Figure 9 polymers-13-02674-f009:**
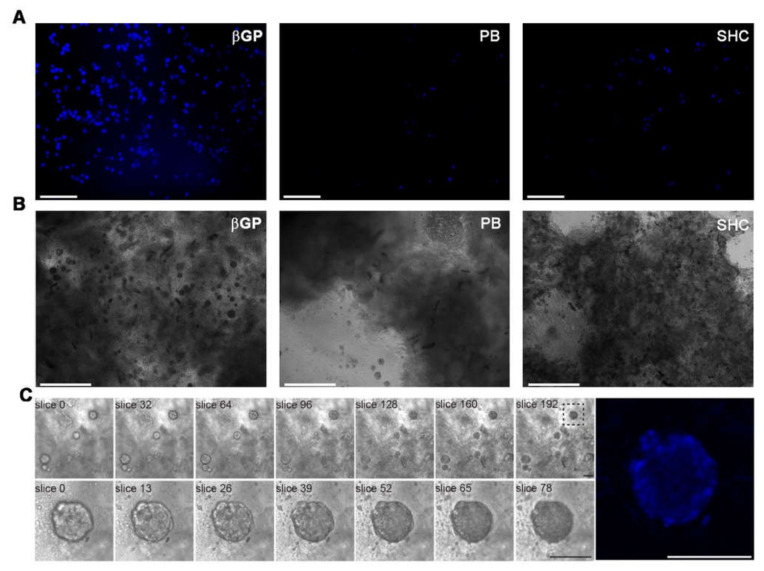
(**A**) Fluorescence microscopy investigation at 24 h of culture of cell-laden Ch-Pec hydrogels formed in the presence of different GAs. (**B**) Optical investigation of spheroids in Ch-Pec hydrogels after 21 days of culture. (**C**) Z-stack analysis at different magnifications of spheroids in Ch-Pec-βGP hydrogels after 21 days of culture (scale bars: 100 µm).

**Table 1 polymers-13-02674-t001:** Initial and final polymer and GA solution concentrations. pH values of Ch and Pec solutions, gelling agent, Ch-Pec mix and final hydrogels. V_i_ = initial volume.

Polymer and Solution Concentrations	Initial Concentration	Initial pH Value	Final pH Value	Final Concentration
Ch in 0.1 M HCl	3.33%	6	-	1.38%
Pec in H20 D.I.	3.33%	5	-	1.38%
Ch-Pec	-	-	6	2.77%
Ch-Pec-βGP (0.04 M)	-	-	7	-
Ch-Pec-βGP (0.08 M)	-	-	7.0/8.0	-
Ch-Pec-βGP (0.16 M)	-	-	7.0/8.0	-
Ch-Pec-PB (0.04 M)	-	-	6.0/7.0	-
Ch-Pec-SHC (0.04 M)	-	-	8	-
Ch-Pec-βGP-DMEM	-	-	7.0/8.0	-
Ch-Pec-PB-DMEM	-	-	7.0/8.0	-
Ch-Pec-SHC-DMEM	-	-	8	-
βGP	0.1 M (V_i_ =1.2 mL)	8.0/9.0	-	0.04 M
βGP	0.2 M (V_i_ =1.2 mL)	-	-	0.08 M
βGP	0.2 M (V_i_ =0.6 mL)	-	-	0.16 M
PB	0.1 M	7	-	0.04 M
SHC	0.1 M	9.0/10.0	-	0.04 M

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
