# Peer review of "Preparation and Characterization of Salt-Mediated Injectable Thermosensitive Chitosan/Pectin Hydrogels for Cell Embedding and Culturing"

_polymers, 2021, doi:10.3390/polym13162674_

Round 1
Reviewer 1 Report
The authors presented a study on the "Preparation and characterization of salt-mediated injectable thermosensitive chitosan/pectin hydrogels for cell-embedding and culturing". The study presents an interesting topic with a well-written manuscript and in my opinion is a good fit for the Polymers journal. I suggest the authors address some minor corrections below to further improve the quality of the manuscript.
(1) Define ECM
(2) Lines 72-77, too long sentence creates ambiguity. Please rephrase.
(3) How did the authors come up with the concentration of Chitosan and pectin (3.33 w/v%) for the hydrogels?
(4) The authors write "solubilizing Ch powder 3.33% (w/v) in aqueous solution of hydrochloric acid (HCl) 0.1 M". Does this mean the authors dissolved chitosan the chitosan in 0.1M HCl solution? or they added 0.1M HCl to facilitate chitosan solution?
(5) The sentence in lines 192-195, "Three weak bases ....... polymer network (semi-IPN)", Too long sentence, please rephrase.
(6) Please rephrase the sentence in lines 344-347 for better clarity.
Author Response
Please, see attachment

Reviewer 2 Report
This is an important manuscript for the readers of this journal, describing the use of a chitosan-pectin hydrogel that can make sol-gel transitions at 37C, in the physiological pH range, potentially of use to support cells as an ECM mimic. However, there are concerns with the technical methodologies used to support this conclusion, that are a cause for concern and dampened enthusiasm. The following major comments should be considered carefully, involving a revision, before re-consideration.
1. Cell encapsulation and support:
- There is no discussion of why some formulations support cell growth versus two others do not.
- Cell encapsulation methods can be outlined better - are cells suspended in complete serum supplemented DMEM, or basal DMEM?
- Evaluating 100ul spots of cell-gel suspension is extremely inadequate and inappropriate to make the claim that the cells are suspended uniformly.
- Furthermore, how thick are the gels? To evaluate nuclear distribution, the authors might consider presenting quantitative evaluation of a z-stack, and not just a low resolution 2D image. These images are not quantiative, and do not support the conclusion that one gel type supports cells while the others do not.
2. Hydrogel characterization:
- Morphological analysis of scanning electron micrographs must be described in the methods, and the quantification of the porosity identified. As such, the interconnectivity of these structures is important to support cells, and must be quantified and reported as part of the main manuscript, and not supplementary figures.
- How was porosity quantified?
3. Mechanical testing
- How were compression tests performed, and why, on hydrogels as opposed to more rheological testing and reporting moduli at 37C. The universal mechanical tester likely does not have a load cell that has the resolution to measure the modulus of a hydrogel accurately.
- In order to report a stable mechanical modulus/property, the authors should consider showing a frequency-independent bulk modulus performed following a stress and strain sweep. This value is likely more representative of the hydrogel mechanical properties rather than the universal mechanical tester.
- An alternate could also be atomic force microscopy.
- Furthermore, comparison to Bombaldi de Souza et al., (2020) in reporting elastic modulus is somewhat questionable, since this manuscript used unconstrained compression testing, while the other used tensile testing.
